# Association of Antibody–Drug Conjugate (ADC) Target Expression and Interstitial Lung Disease (ILD) in Non-Small-Cell Lung Cancer (NSCLC): Association or Causation or Neither?

**DOI:** 10.3390/cancers16223753

**Published:** 2024-11-07

**Authors:** Aakash Desai, Vivek Subbiah, Sinchita Roy-Chowdhuri, Ajay Sheshadri, Sameer Deshmukh, Solange Peters

**Affiliations:** 1Department of Medicine, University of Alabama, Birmingham, AL 35294, USA; ssdeshmukh@uabmc.edu; 2Sarah Cannon Research Institute, Nashville, TN 37203, USA; vivek.subbiah@scri.com; 3Department of Pathology, The University of Texas MD Anderson Cancer Center, Houston, TX 77030, USA; sroy2@mdanderson.org; 4Department of Pulmonary Medicine, The University of Texas MD Anderson Cancer Center, Houston, TX 77030, USA; asheshadri@mdanderson.org; 5Department of Medical Oncology, Lausanne University, 1015 Lausanne, Switzerland; solange.peters@chuv.ch

**Keywords:** antibody–drug conjugates, ADCs, interstitial lung disease, non-small-cell lung cancer, NSCLC, toxicity, ILD, TROP2, ERBB2

## Abstract

Antibody Drug Conjugates (ADCs) are a newer class of therapeutic agents which present a promising therapeutic approach for non-small cell lung cancer (NSCLC). Several ADCs are in clinical trial, targeting various antigens including TROP2, HER2, MET and others. Despite the promise, ADCs present unique challenges including toxicity like interstitial lung disease (ILD). A more thorough analysis of target expression could lead to a better understanding of adverse events, such as ILD. This improved understanding may help reduce the occurrence of these adverse effects. Our study aims to analyze the ADC target expression in lung tissue to help us understand the relation between treatment with ADCs and onset of interstitial lung disease.

## 1. Introduction

Non-small-cell lung cancer (NSCLC) is the most commonly encountered type of lung cancer worldwide [1]. Over the last decade, the advent of immune checkpoint inhibitors and genomically targeted therapy has dramatically improved the outcomes of patients with NSCLC [2,3]. However, mortality remains high in patients with NSCLC, and significant room for improvement remains; novel therapies with new mechanisms of action are still needed [4]. One promising approach for treatment of NSCLC is using antibody–drug conjugates (ADCs) to target specific tumor antigens. ADCs are monoclonal antibodies linked to cytotoxic drugs, which theoretically allow for the selective delivery of cytotoxic agents to cancer cells while avoiding harm to normal healthy cells and tissues [5].

The clinical development of ADCs has not been without challenge since the first clinically approved ADC, gemtuzumab ozogamicin, in hematological malignancies. More recently, improved ADC designs have led to several novel ADCs receiving FDA approval for gastric cancer, breast cancer, and, more recently, NSCLC [6,7]. The success of ADCs in NSCLC was demonstrated in the DESTINY-Lung 01 trial evaluating trastuzumab deruxtecan (TDX-d) in patients with metastatic, HER2-mutant NSCLC [8]. In 91 patients, TDX-d administration resulted in an objective response rate (ORR) of 55% with a median duration of response (mDOR) of 9.3 months. However, 26% of the patients in that study had any-grade drug-induced interstitial lung disease (D-ILD), using modestly higher doses of TDX-d as compared to breast cancer (6.4 vs. 5.4 mg/kg). Studies of anti-ERBB2 ADCs have also shown that D-ILD is a common dose-limiting toxicity, and the highest incidence of any-grade D-ILD was in patients with lung cancer [9]. The early and accurate diagnosis of D-ILD remains a significant challenge in the clinic because D-ILD can often mimic infectious pneumonia or cancer progression. Improving our understanding of this unique toxicity will help clinicians improve patient outcomes in those receiving ADCs for NSCLC.

Risk factors for D-ILD associated with ADCs may vary based upon cytotoxic payload, mechanism of action, underlying patient comorbidities, and the type of cancer. Patients with NSCLC may be at a higher risk for D-ILD given the association of NSCLC with smoking history and COPD, as well as lung surgery and thoracic radiation. With the ongoing development of multiple ADCs for use in NSCLC, developing a deeper understanding of the mechanism of ADC-induced ILD will enable the optimization of ADC development. A meta-analysis encompassed a comprehensive evaluation of 39 studies involving 7732 patients, to ascertain the incidence of market-approved ADC-related pneumonitis in solid tumors. The findings reveal that ADCs are associated with a significant incidence of all grade and grade ≥3 pneumonitis, with the highest incidence being observed in NSCLC 22.18% (95% CI, 2.14–52.61%) [10]. As such, interstitial lung disease (ILD) constitutes a significant challenge in the context of ADC use in lung cancer.

Beyond the mechanism of action of ADCs and the additional factors described above, ADC target expression could also impact ILD toxicity. In this review, we study the landscape of ADC target expression at the RNA, single-cell RNA, and protein level in the normal lung. We also review the incidence of ILD in clinical studies with ADCs to infer a potential relationship between ADC target antigen expression in the lungs and D-ILD.

## 2. Methods

A. Identification of targets: A systematic search of the MEDLINE (Ovid), Embase (Elsevier), and CENTRAL (Cochrane Library) databases, as well as peer-reviewed literature in the English language, conference abstracts, and trial registrations from major international oncology meetings up to March 2023 were conducted. The search was restricted to studies written in or translated into English and discussed keywords such as “Antibody Drug Conjugates”, “non-small cell lung cancer”, “NSCLC”, “ADC”, and “lung adenocarcinoma”. Clinical trials describing ADC-based interventions in NSCLC were identified through a review of each title and abstract. All ADCs regardless of their FDA approval status were studied. Targets of interest were then identified based on this information.

B. Incidence of ILD: The publicly available trial results of phase 1/2 clinical trials for each of the ADC corresponding to the target antigens of interest were included. Studies of trastuzumab deruxetecan (ERBB2) [8], patritumab deruxetecan (ERBB3) [11], datopotamab deruxetecan (TROP2) [12], tusamitamab ravtansine (CEACAM5) [13], and Telisotuzumab Vedotin (MET) [14] were reviewed to extract data on the incidence of ILD. We also reviewed the FDA label and package inserts for FDA-approved agents.

C. RNA and protein expression: We used publicly available datasets to study bulk RNA, single-cell RNA and protein expression via IHC of target antigens of interest for ADCs. The Genotype-Tissue Expression (GTEx) project is an ongoing effort to build a comprehensive public resource to study tissue-specific gene expression and regulation. These data were used to conduct the bulk RNA-seq analysis [15]. Data from The Cancer Genome Atlas (TCGA) were used to conduct the bulk RNA-seq analysis to understand RNA level expression of genes of interest [16]. The Human Protein Atlas (HPA) was used to obtain single-cell RNA sequencing (scRNA-seq) data and protein expression based on conventional immunohistochemistry profiling [17]. The HPA’s scRNA-seq data relies on the Smart-seq2 protocol, involving individual cell isolation from tissue samples, RNA reverse transcription, and amplification from each cell. Expression data are normalized using the transcripts per million (TPM) approach. Normalized mean transcript expression, represented as nTPM, serves as the expression-level measure for each target antigen. Detailed methods on how these data were obtained for the HPA database have been described elsewhere [18]. Protein expression data were available as antibody staining in the annotated cell types in the lung tissue reported as not detected, low, medium, or high, based on conventional immunohistochemistry profiling. This score was based on the combination of staining intensity and fraction of stained cells. Furthermore, staining intensity data were extracted from representative samples and scored as follows: 0—negative; 1—weak; 2—moderate; and 3—high. We used the highest antibody staining values for when multiple antibody staining patterns were described.

## 3. Results

We found that, based on bulk RNA expression, these antigen targets of interest had higher expression in tumor tissues compared to normal lung tissue (obtained from GTEx and adjusted normal lung tissue from TCGA) (Figure 1) (Table 1).

We evaluated single-cell RNA sequencing data from normal lungs and found that all target antigens were expressed in lung tissues, with the highest normalized transcript expression levels found in alveolar cells (type 1 or type 2). The nTPM expression for single-cell RNA across alveolar cells in lung tissue ranged from 73.7 to 1549.8 (Figure 2).

On analysis of protein expression from HPA via IHC, three of the five targets (ERBB2, ERBB3, and CEACAM5) demonstrated moderate-intensity staining (2+) on IHC, while MET demonstrated weak staining on alveolar cells. Similarly, IHC staining was moderate-to-high-intensity for macrophages in ERBB2, ERBB3, CEACAM5, and MET. Representative images for IHC antibody staining obtained directly from the HPA are shown in Figure 3.

Our analysis revealed that the incidence of all-grade D-ILD across the ADCs of interest ranged from 0 to 26.4% (Table 1). Furthermore, there was no clear pattern of the incidence of ILD with IHC staining intensity across targets of interest, indicating that D-ILD may occur irrespective of the expression of proteins of interest and may in turn be related to linker and cytotoxic payload characteristics rather than target antigen expression.

## 4. Discussion

Beyond immune checkpoint inhibitors and genomically targeted therapy, ADCs are poised to enlarge the landscape of NSCLC therapy in the near future. It is important to understand the mechanisms of D-ILD. Our study identified that most of the ADC-targetable antigens were expressed in normal lung tissues, with the highest normalized transcript expression levels for these found within alveolar cells. We found that despite moderate-to-high-intensity IHC staining on alveolar cells, bulk or scRNA-seq expression of ADC-targetable antigens does not explain the incidence of any-grade ILD. Actually, there was no clear pattern of the incidence of ILD with IHC staining intensity across targets of interest, indicating that D-ILD may occur irrespective of the expression of proteins of interest. Although alveolar type 1 and type 2 cells most commonly expressed the targets of interest, single-cell RNA expression of these target genes were not proportionally associated with the occurrence of D-ILD. For example, the expression of TROP2 on alveolar type 1 cells exceeded ERBB2 expression (1549.8 vs. 73.7 nmTP); however, the incidence of ILD with datopotamab deruxtecan was found to be much lower that the results of DESTINY-lung 01 for trastuzumab deruxtecan (11 vs. 26%).

From a pathologic standpoint, D-ILD encompasses various lung conditions characterized by alveolitis, disarray in alveolar structures, and fibrosis in the alveolar interstitium. These processes culminate in a compromised gas exchange. The onset of this cascade is triggered by an acute or chronic lung injury caused by a drug, which subsequently advances to inflammation and manifests as overt D-ILD [19]. Many chemotherapeutic drugs, such as bleomycin, taxanes, irinotecan, tyrosine kinase inhibitors, and immune checkpoint inhibitors, have been associated with various forms of D-ILD [20]. Of interest, irinotecan, a topoisomerase I inhibitor, causes D-ILD, with higher incidence in patients with lung cancer specifically [21]. Many ADCs currently being developed in the clinic include a deruxtecan payload, which also acts via inhibition of topoisomerase I. Interestingly, in our review, we found that the incidence of ILD was higher with ADCs that have a DXd payload (Table 1). Although the specific mechanism of injury induced by ADCs is yet to be elucidated, it is possibly associated, at least in part, with the carried payload, with a low contribution of target antigen expression in lung tissue. For example, interstitial lung disease has been identified as a known adverse event for topotecan, a topoisomerase inhibitor [22,23].

Our results are indicative of D-ILD likely being an off-target, on-tissue toxicity of ADCs. It has been previously suggested that ADC-induced lung injury may be associated with bystander killing by free payload released from cancer cells, as well as circulating free payload resulting from deconjugation of the ADC. Furthermore, preclinical studies suggest that ADC-induced alveolar damage could be related to target-independent uptake of the conjugate by immune cells, rather than target-dependent uptake [24]. This further confirms our findings of a target-independent origin of ILD toxicity.

Finally, the type of linker and drug-to-antibody ratio used in the ADC construct may have important implications for the degree of bystander anti-tumor effect and off-target on-tissue toxicity. For example, the incidence of ILD was much higher in patients with HER2-positive breast cancer when treated with TDX-d compared with trastuzumab emtansine (28% vs. 15%), which may be related to the presence of a tumor-selective cleavable linker and a higher drug-to-antibody ratio in TDX-d [25,26]. Of note, TDX-d lung toxicity was shown to be strongly dose-dependent, arguing for the relevance of the level of exposure of normal lung tissue to the cytotoxic payload. Cleavable linkers were designed to improve ADC efficacy, as they can break down and release the cytotoxic payload in response to specific tumor-associated factors, such as acidic or reducing conditions, or proteolytic enzymes. On the other hand, non-cleavable linkers are more stable in the bloodstream but rely on the entire antibody–linker complex being degraded in lysosomes to release their payloads [27]. It is worth noting that while the extracellular release of the cytotoxic payload could be a crucial factor in the success of ADC therapy, this may have importance in the degree of on-target on-tissue toxicity observed [28]. Therefore, finding the right balance of linker stability is a complex challenge that depends on the specific target and payload selected, as well as the unique characteristics of the tumor microenvironment.

Optimization of ADCs should focus on improving activity in parallel to reducing bystander effects by selecting linker and payload classes that are least likely to cause ILD. This optimization should also take into account the dose and drug-to-antibody ratio. Our study supports this recommendation by identifying that ILD may be a bystander effect and highlighting the need to optimize ADCs to reduce such effects.

The limitations of our study include the use of scRNA-seq data from a single database and the limited sample size of clinical trial data for ILD incidence, given that many of these studies are currently in phase 1 and/or 2. Further studies using larger sample sizes and prospective IHC staining, bulk RNA sequencing, and scRNA-seq of tissue biopsies may provide a more comprehensive insight into the expression of these target antigens in normal and tumor lung tissue and the mechanism of ILD toxicity.

## 5. Conclusions

In conclusion, our study provides valuable insights into the incidence of all-grade ILD across different ADCs currently studied in NSCLC. Furthermore, there did not seem to be any correlation between the expression of target antigens and the incidence of ILD. This highlights the importance of well-designed translational and correlative studies to inform the mechanisms behind ADC-induced ILD. While it is important to promptly recognize and avoid inciting agents, there is a need to test effective therapies for ADC-induced ILD. Ultimately, better understanding of the pathogenesis of ADC-induced ILD will enable improved drug development of the ADCs currently in the pipeline while balancing bystander effects with target selectivity.

## Figures and Tables

**Figure 1 cancers-16-03753-f001:**
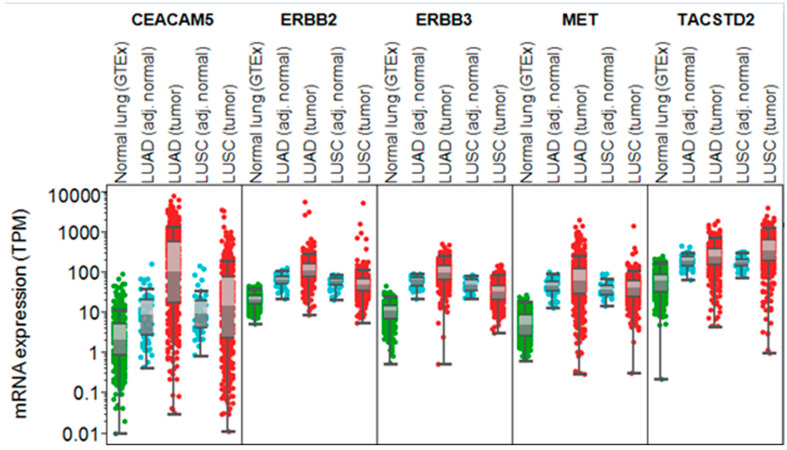
TCGA and GTEx bulk RNA expression of ADC-targetable antigens. Note: mRNA expression for the different targets in normal lung tissue was obtained from GTEx, while expression in lung adenocarcinoma (LUAD) and squamous carcinomas (LUSCs), as well as their adjacent non-cancerous tissue, was extracted from TCGA.

**Figure 2 cancers-16-03753-f002:**
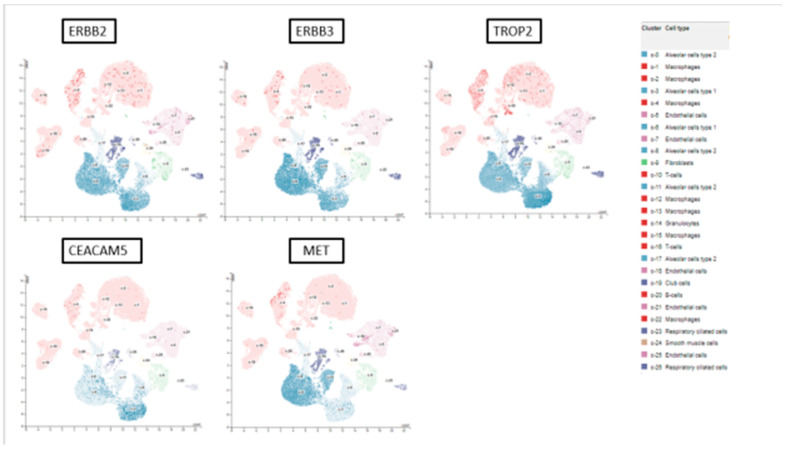
scRNA-seq expression of ADC-targetable antigens in lung tissue (details: https://www.proteinatlas.org/ENSG00000141736-ERBB2/single+cell, accessed on 29 October 2024).

**Figure 3 cancers-16-03753-f003:**
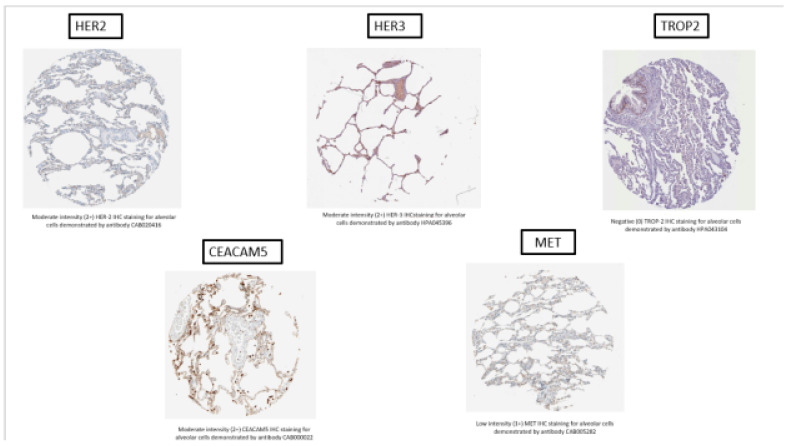
Representative IHC staining of normal lung tissue for HER2, HER3, TROP2, CEACAM5, and MET target antigens from the Human Protein Atlas (details: https://www.proteinatlas.org/ENSG00000141736-ERBB2/tissue/lung, accessed on 29 October 2024).

**Table 1 cancers-16-03753-t001:** Characteristics of ADC, ILD incidence, and target antigen expression.

ADC	Target	No. of Patients Included in Study	% All-Grade ILD	nMTP Expression	scRNA-seq Cell Type with the Highest Expression	Payload	Linker	DAR	Reference
Trastuzumab Deruxetecan	HER2	91	26.4	73.7	Type 1	Topoisomerase 1 inhibitor (Dxd)	Cleavable	8	[8]
Patritumab Deruxetecan	HER3	57	7	88.5	Type 2	Topoisomerase 1 inhibitor (Dxd)	Cleavable	8	[11]
Datopotamab Deruxetecan	TROP2	34	3	1549.8	Type 1	Topoisomerase 1 inhibitor (Dxd)	Cleavable	4	[12]
Tusamitamab Ravtansine	CEACAM5	24	0	174.4	Type 1	Microtubule inhibitor (DM4)	Cleavable	3.8	[13]
Telisotuzumab Vedotin	MET	136	6.6	155	Type 2	Microtubule inhibitor (MMAE)	Cleavable	3	[14]

## Data Availability

The data can be shared up on request.

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
