# Peer review of "Association of Antibody–Drug Conjugate (ADC) Target Expression and Interstitial Lung Disease (ILD) in Non-Small-Cell Lung Cancer (NSCLC): Association or Causation or Neither?"

_cancers, 2024, doi:10.3390/cancers16223753_

Round 1
Reviewer 1 Report
Comments and Suggestions for Authors
This is a detailed and comprehensive review. The study focuses on the potential of antibody-drug conjugates (ADCs) in the treatment of non-small cell lung cancer (NSCLC) and the mechanisms underlying ADC-induced interstitial lung disease (D-ILD), which is one of the major challenges for ADC therapy. However, there seem to be several areas that could benefit from deeper exploration.
The study investigates the relationship between target antigen expression and the occurrence of D-ILD using RNA and protein expression data, but it reports that no clear correlation was observed. Has the study sufficiently considered the limitations of expression data and technical challenges that could explain the lack of correlation? Specifically, how were potential issues regarding the reliability of the data, such as small sample sizes in scRNA-seq data or tissue heterogeneity, addressed? The reliance on scRNA-seq data and IHC staining with limited sample sizes or dependence on a single database raises concerns about whether these factors might impact the study’s conclusions.
The possibility that D-ILD could occur "independently of target antigen expression" has been suggested, which could imply that the traditional mechanism of ADCs (selective delivery to cancer cells) may not be functioning as expected. Could this point to issues in the design or stability of the ADC itself? For example, how do you interpret the possibility that the optimization of linker type and drug-to-antibody ratio (DAR) remains inadequate? If factors related to linker type or drug stability are not fully optimized, there is a concern that the selective toxicity may not be fully realized, increasing the risk of off-target effects.
The study mentions the possibility that the "bystander effect" could contribute to the development of D-ILD. What experiments or additional data do you believe would be necessary to support this hypothesis? Specifically, there seems to be a lack of detailed planning on methodologies or strategies to validate the bystander effect in preclinical models. What are your thoughts on this?
The risk of D-ILD in NSCLC patients is highlighted as being particularly high, but there is little mention of the availability of existing clinical prediction models or biomarkers that could be used to quantitatively assess this risk. What new biomarkers or assessment criteria do you believe should be developed for the early diagnosis or prevention of D-ILD? Additionally, how do you think the currently available predictive models can be improved?
Regarding the diagnosis and mechanisms of D-ILD, the comparison with toxicity from other ADCs does not seem to be deeply discussed. For instance, while the high incidence of D-ILD with ADCs containing the topoisomerase I inhibitor DXd is mentioned, how do you believe the mechanisms differ when compared to ADCs utilizing other drugs or linkers? Furthermore, what lessons can be learned from the toxicity profiles of existing topoisomerase I inhibitors?
Author Response
We are thankful to the reviewers for their review and feedback of our manuscript. We believe the recommendations will make our manuscript more robust and informative to the broader oncology community.
Reviewer 1
The study investigates the relationship between target antigen expression and the occurrence of D-ILD using RNA and protein expression data, but it reports that no clear correlation was observed. Has the study sufficiently considered the limitations of expression data and technical challenges that could explain the lack of correlation? Specifically, how were potential issues regarding the reliability of the data, such as small sample sizes in scRNA-seq data or tissue heterogeneity, addressed? The reliance on scRNA-seq data and IHC staining with limited sample sizes or dependence on a single database raises concerns about whether these factors might impact the study’s conclusions.
Response: We thank the reviewers for their comment. We completely agree that there are certain limitations with expression data and small sample size. We do acknowledge this in the discussion section where we have explicitly stated “The limitations of our study include the use of scRNA-seq data from a single database and the limited sample size of clinical trial data for ILD incidence given many of these studies are currently in phase 1 and/or 2. Further studies using larger sample sizes and prospective IHC staining, bulk RNA sequencing, and scRNA-seq of tissue biopsies may provide a more comprehensive insight into the expression of these target antigens in normal and tumor lung tissue, and the mechanism of ILD toxicity.”
The possibility that D-ILD could occur "independently of target antigen expression" has been suggested, which could imply that the traditional mechanism of ADCs (selective delivery to cancer cells) may not be functioning as expected. Could this point to issues in the design or stability of the ADC itself? For example, how do you interpret the possibility that the optimization of linker type and drug-to-antibody ratio (DAR) remains inadequate? If factors related to linker type or drug stability are not fully optimized, there is a concern that the selective toxicity may not be fully realized, increasing the risk of off-target effects.
Response: We thank the reviewers for their comment. We agree that optimization of Linker type and drug to antibody ratio is of paramount importance. We state in the discussion section the following: “Optimization of ADCs should focus on improving activity in parallel to reducing bystander effects by selecting linker and payload classes that are least likely to cause ILD. This optimization should also consider the dose and drug -to antibody ratio. Our study supports this recommendation by identifying that ILD may be a bystander effect and highlighting the need to optimize ADCs to reduce such effects.”
The study mentions the possibility that the "bystander effect" could contribute to the development of D-ILD. What experiments or additional data do you believe would be necessary to support this hypothesis? Specifically, there seems to be a lack of detailed planning on methodologies or strategies to validate the bystander effect in preclinical models. What are your thoughts on this?
Response: We thank the reviewers for their comment. We agree that more research is needed in understanding the true value of bystander effect both from efficacy and safety standpoint. Given our focus was more on clinical implications of this, we did not comment specifically on methodologies and strategies to validate this in preclinical models.
The risk of D-ILD in NSCLC patients is highlighted as being particularly high, but there is little mention of the availability of existing clinical prediction models or biomarkers that could be used to quantitatively assess this risk. What new biomarkers or assessment criteria do you believe should be developed for the early diagnosis or prevention of D-ILD? Additionally, how do you think the currently available predictive models can be improved?
Response: We thank the reviewers for their comment. We agree that more research is needed in understanding biomarkers and clinical predictive variables to quantitatively assess the risk of drug-induced ILD. Given our focus was more on clinical prevalence and correlation with target expression, we did not specifically look into any predictive modeling or biomarkers.
Regarding the diagnosis and mechanisms of D-ILD, the comparison with toxicity from other ADCs does not seem to be deeply discussed. For instance, while the high incidence of D-ILD with ADCs containing the topoisomerase I inhibitor DXd is mentioned, how do you believe the mechanisms differ when compared to ADCs utilizing other drugs or linkers? Furthermore, what lessons can be learned from the toxicity profiles of existing topoisomerase I inhibitors?
Response: We thank the reviewers for their comment. We have added on page 7, where the paragraph now reads: “Many ADCs currently being developed in the clinic include a deruxtecan payload, which also acts via inhibition of topoisomerase I. Interestingly, in our review, we found that the incidence of ILD was higher with ADCs having a DXd payload (Table 1). Although the specific mechanism of injury induced by ADCs is yet to be elucidated, it is possibly associated, at least in part, with the carried payload with little contribution of target antigen expression in lung tissue. For example, interstitial lung disease has been identified as a known adverse event for topotecan, a topoisomerase inhibitor.”
We have also included references 22, 23 for the reader for in depth review of this phenomenon with topoisomerase inhibitors.
Reviewer 2 Report
Comments and Suggestions for Authors
Interesting MS on a subject ─ ILD induced by the usage of ADCs in NSCLC patients ─ that is very relevant for optimizing the future application of this promising type of targeted treatment for NSCLC. However, the presentation of data needs to be clearer and more extensive before the MS is acceptable for publication.
SPECIFIC POINTS
Methods
Part A. Identification of targets, line 98-99, “Targets of interest were then identified based on this information, as presented in Table 1”. The table belongs to the Results section and should be described there, as it contains several relevant results. In the current MS, Table 1 (contains data) is only mentioned in the Methods and in the Discussion, but not described in the Results, which seems weird.
Part C. RNA and protein expression, line 106-108, “The Genotype-Tissue Expression (GTEx) project is an ongoing effort to build a comprehensive public resource to study tissue-specific gene expression and regulation.[15]” The sentence is a description of what GTEx is but does not explain how the authors have used it for extracting data for their study.
Line 108-110, “The Cancer Genome Atlas (TCGA) data was used to conduct the bulk RNA-seq analysis to understand RNA level expression of genes of interest.[16]”: It seems as if the authors just used the RNA data available in TCGA rather than “conducting the bulk RNA-seq analysis” by themselves. A similar approach was used for extracting scRNA-seq and IHC data from the HPA, while no “home-made” data generated by the authors themselves were used.
Line 118-119, “… data were available as antibody staining in the annotated cell types in the lung tissue reported as not detected, low …”: presumably “as” before “reported” should be used to make the sentence clearer.
Line 123-124, “We used the highest antibody staining values for when multiple antibodies staining patterns were described”: these values seem to be missing as they are not shown in the Results or elsewhere in the MS.
Results
Figure 1: For clarity, please explain in the figure legend and possibly in the text that TACSTD2 is the tumor-associated calcium signal transducer 2 gene that codes for TROP2. The whole figure should also be explained better in the legend, mentioning that mRNA expression for the different targets in normal lung tissue was obtained from GTEx, while expression in lung adenocarcinoma (LUAD) and squamous carcinomas (LUSC) as well as their adjacent non-cancerous tissue was extracted from TCGA.
Further regarding Figure 1: Are the difference in mRNA expression significant for each gene transcript?
Figure 2: in the version submitted for review, is a very out-of-focus figure. It is very difficult to see the names of cell types and impossible to see the numbers associated with the sc types. The units on x and y axis are not visible either. To evaluate the figure 2, the reviewer had to look up in HPA and find the five figures related to the scRNA expression for the five ADC-targetable antigens. Although, this is feasible, it is not acceptable by future readers of the MS. Thus, a much improved (sharp) version of the figure should be used for publication.
Line 132-133, “… the highest normalized transcript expression levels found in alveolar cells (type 1 or type 2).”: It is unclear how the authors came to this conclusion, as the highest expression appears to be in macrophages. Could this be marked on figure 2 for clarity?
Figure 3: Same problem as for fig.2, it’s not sharp at all, so that it is very difficult/impossible to see the immunostainings and to read the text under each of the five pictures from the HPA. A sharper version with much higher resolution should be provided (again, the evaluation of the claimed results had to be done by this reviewer on the original HPA website …).
Line 137-139, “HPA, three of the five targets (ERBB2, ERBB3, CEACAM5) demonstrated moderate intensity staining (2+) on IHC, while MET demonstrated weak staining on alveolar cells.”: for completeness and clarity, the expression of TROP2 shown on fig. 3 (and corresponding to none in the HPA website) should also be described.
The HPA also reports the expression in bronchi/bronchioles, which in principle might be relevant for the study, as respiratory symptoms can also derive from bronchitis or bronchiolitis/small airway diseases.
Table 1: its formatting could be improved (for ex., abundant space is dedicated to the column with single references, while the text in several of the other narrower columns is “broken”). Furthermore, the name of each study/trial should be shown in the table and the abbreviation DAR should be defined.
Discussion
Line 157-159, “We found that despite moderate to high intensity IHC staining on alveolar cells, … does not explain the incidence of any-grade ILD”: This is somehow confusing, as for instance MET protein assessed by IHC is shown in the HPA (and described in the Results by the authors) to be weak in alveolar cells.
The most meaningful correlation protein expression vs ILD would be to see the expression of targets in biopsies from pts with ILD vs pts w/o ILD, which is not feasible for the current study. Yet, it should be discussed that a limitation of study is extrapolating the expression in normal lung tissue in the HPA as representing the expression of pts with ILD, w/o considering the interindividual expression and the effect that inflammation can cause on the expression of the relevant target proteins.
The authors claim that “D-ILD may occur irrespective of the expression of proteins of interest”. However, they base this conclusion on the expression of mRNA which seems strange as ultimately the target for ADCs are antigens (proteins), not mRNAs.
In this respect, the authors write that “For example, expression of TROP2 on alveolar type 1 cells exceed that of ERBB2 expression (1549.8 vs 73.7 nmTP), however the incidence of ILD with datopotamab deruxtecan was found to much lower that results from DESTINY-lung 01 for trastuzumab deruxtecan (11 vs 26%)”. The issue is confusing, as the HPA shows that TROP2 protein expression is absent in alveolar cells and macrophages (with any of the tested Abs). Thus, if the protein is not there, why should the transcript be associated with ADC-induced ILD?
Line 202-203, “this may have importance in the degree of on-target on-tissue toxicity observed”: should it be “on-target” or “off-target” here?
It is worth mentioning, at least in the Discussion, that the bispecific anti EGFR-MET Ab, amivantamab, has also been associated with ILD/Pneumonitis when used alone or in combination with lazertinib in pts. with NSCLC. Even the company selling amivantamab mentions that amivantamab in combination with lazertinib can cause severe and fatal ILD/pneumonitis, which in the MARIPOSA trial occurred in 3.1% of pts. treated with this combination, including Grade 3 in 1.0% and Grade 4 in 0.2% of pts. In that study, there was 1 fatal case of ILD/pneumonitis and the combined treatment was permanently discontinued in 2.9% of pts. because of ILD/pneumonitis.
Conclusion
Line 220-221, “there did not seem to be any correlation between the expression of target antigens and the incidence of ILD”: it would seem appropriate to state that this was a correlation with the general normal expression as assessed by databases.
Line 223-224, “While it is important to promptly recognize and avoid inciting agents”: what do the authors mean by that?
As minor points, some small adjustments in the text are needed, for instance:
- space before Introduction, Methods, Discussion, Conclusion in the Abstract;
- the abbreviation ADCs sometimes written as ADC’s when it’s not genitive;
- “A systemic search … were conducted” (line 90-93);
- “was found to much lower that results from …” (line 166)
- Some places HER2 and HER3 (older nomenclature) are used, other places ERBB2 and ERBB3. It would be more appropriate to be consistent and use the same terms (for ex., either HER2 and HER3 or ERBB2 and ERBB3) throughout the MS. Alternatively both (HER2/ERBB2, HER3/ERBB3).
Author Response
Reviewer 2
Methods
Part A. Identification of targets, line 98-99, “Targets of interest were then identified based on this information, as presented in Table 1”. The table belongs to the Results section and should be described there, as it contains several relevant results. In the current MS, Table 1 (contains data) is only mentioned in the Methods and in the Discussion, but not described in the Results, which seems weird.
Response: We thank the reviewers for their comment. We have made this correction and moved the table 1 citation to the results section.
Part C. RNA and protein expression, line 106-108, “The Genotype-Tissue Expression (GTEx) project is an ongoing effort to build a comprehensive public resource to study tissue-specific gene expression and regulation.[15]” The sentence is a description of what GTEx is but does not explain how the authors have used it for extracting data for their study.
Response: We thank the reviewers for their comment. We have added: “The Genotype-Tissue Expression (GTEx) project is an ongoing effort to build a comprehensive public resource to study tissue-specific gene expression and regulation. This data was used to conduct the bulk RNA-seq analysis”
Line 108-110, “The Cancer Genome Atlas (TCGA) data was used to conduct the bulk RNA-seq analysis to understand RNA level expression of genes of interest.[16]”: It seems as if the authors just used the RNA data available in TCGA rather than “conducting the bulk RNA-seq analysis” by themselves. A similar approach was used for extracting scRNA-seq and IHC data from the HPA, while no “home-made” data generated by the authors themselves were used.
Line 118-119, “... data were available as antibody staining in the annotated cell types in the lung tissue reported as not detected, low ...”: presumably “as” before “reported” should be used to make the sentence clearer.
Line 123-124, “We used the highest antibody staining values for when multiple antibodies staining patterns were described”: these values seem to be missing as they are not shown in the Results or elsewhere in the MS.
Response: We thank the reviewers for their comment. This is a publicly available dataset which has been used to conduct the analysis. We have stated this with adding this sentence: “RNA and protein expression: We used publicly available datasets to study bulk RNA, single-cell RNA and protein expression via IHC of target antigens of interest for ADCs.” No inhouse analysis or sequencing for done for this project, neither this is claimed by authors.
Results
Figure 1: For clarity, please explain in the figure legend and possibly in the text that TACSTD2 is the tumor-associated calcium signal transducer 2 gene that codes for TROP2. The whole figure should also be explained better in the legend, mentioning that mRNA expression for the different targets in normal lung tissue was obtained from GTEx, while expression in lung adenocarcinoma (LUAD) and squamous carcinomas (LUSC) as well as their adjacent non-cancerous tissue was extracted from TCGA. Further regarding Figure 1: Are the difference in mRNA expression significant for each gene transcript?
Response: We thank the reviewers for their comment. We have now added a legend to figure 1: “mRNA expression for the different targets in normal lung tissue was obtained from GTEx, while expression in lung adenocarcinoma (LUAD) and squamous carcinomas (LUSC) as well as their adjacent non-cancerous tissue was extracted from TCGA. “ As is seen from the figure- no clear differences of significance were seen for each gene transcript.
Figure 2: in the version submitted for review, is a very out-of-focus figure. It is very difficult to see the names of cell types and impossible to see the numbers associated with the sc types. The units on x and y axis are not visible either. To evaluate the figure 2, the reviewer had to look up in HPA and find the five figures related to the scRNA expression for the five ADC-targetable antigens. Although, this is feasible, it is not acceptable by future read ers of the MS. Thus, a much improved (sharp) version of the figure should be used for publication.
Response: We thank the reviewers for their comment. We are able to provide a detailed figure for each target which can be included in the supplements along with hyperlinks for the reader to view in detail.
Round 2
Reviewer 1 Report
Comments and Suggestions for Authors
The authors have adequately addressed all of the reviewer’s comments. Thank you for your efforts on this.